# Evaluation of the Degree of Conversion, Residual Monomers and Mechanical Properties of Some Light-Cured Dental Resin Composites

**DOI:** 10.3390/ma12132109

**Published:** 2019-06-30

**Authors:** Marioara Moldovan, Robert Balazsi, Andrada Soanca, Alexandra Roman, Codruta Sarosi, Doina Prodan, Mihaela Vlassa, Ileana Cojocaru, Vicentiu Saceleanu, Ioan Cristescu

**Affiliations:** 1Department of Polymer Composites, Babes-Bolyai University, Institute of Chemistry Raluca Ripan, 30 Fantanele Str., 400294 Cluj-Napoca, Romania; 2Department of Psychology, Babes-Bolyai University, 37 Republicii Street, 400015 Cluj-Napoca, Romania; 3Department of Periodontology, Iuliu Haţieganu University of Medicine and Pharmacy, 15 V. Babes St., 400012 Cluj-Napoca, Romania; 4Department of Horticulture and Food Science, University of Craiova, 13 Al. I. Cuza Str., 200585 Craiova, Romania; 5Faculty of Medicine, University Lucian Blaga Sibiu, 2A Lucian Blaga Str., 550169 Sibiu, Romania; 6Department Orthoped & Traumatol, Clinical Emergency Hospital Bucharest, 8 Floreasca Ave, Sect 1, 014461 Bucharest, Romania

**Keywords:** composites, mechanical properties, degree of conversion, FTIR

## Abstract

The novelty of this study consists in the formulation and characterization of three experimental dental composites (PM, P14M, P2S) for cervical dental lesion restoration compared to the commercial composites Enamel plus HRi® - En (Micerium S.p.A, Avengo, Ge, Italy), G-ænial Anterior® - Ge, (GC Europe N.V., Leuven, Belgium), Charisma® - Ch (Heraeus Kulzer, Berkshire, UK). The physio-chemical properties were studied, like the degree of conversion and the residual monomers in cured samples using FTIR-ATR (attenuated total reflectance) and HPLC-UV (ultraviolet detection), as well as the evaluation of the mechanical properties of the materials. The null hypothesis was that there would be no differences between experimental and commercial resin composites regarding the evaluated parameters. Statistical analysis revealed that water and saliva storage induced significant modifications of all mechanical parameters after three months for all tested materials, except for a few comparisons for each type of material. Storage medium seemed not to alter the values of mechanical parameters in comparison with the initial ones for: diametral tensile strength (DTS-saliva for Ge and PM, compressive strength (CS)-water for Ch, DTS-water and Young’s modulus YM-saliva for P14M and YM-water/ saliva for P2S (*p* > 0.05). Two of the experimental materials showed less than 1% residual monomers, which sustains good polymerization efficiency. Experimental resin composites have good mechanical properties, which makes them recommendable for the successful use in load-bearing surfaces of posterior teeth.

## 1. Introduction

In recent decades, resin composites have been proposed as an alternative to amalgam, gold and ceramic restorations because of their aesthetic and toxicological considerations [1,2]. 

From a clinical point of view, posterior teeth restoration composites require good mechanical properties, such as a high viscosity of the composites and reduced polymerization shrinkage to avoid the degradation and fractured restoration marginal.

The variation of forces in the case of posterior composites can be explained by differences, such as the chemical composition (of the organic phase and inorganic phase), distribution and size of the inorganic particles. [3].

The higher is the fillers volume, the higher is the surface hardness and the compressive strength, and also the increase of the elastic modules, which reduces the flowing ability [4].

Studies show that this does not compensate the polymerization shrinkage, thus leading to high intrinsic stress, but there is a positive correlation between diametrical tensile strength and compressive strength [5].

A restorative composite with high, mechanical properties should be resistant to masticatory forces in clinical situations [6]. Scientific manipulation of resin composite composition and improvements in manufacturing technologies could enhance the mechanical properties and stability of the restorations.

Dental resin composites are complex mixtures containing organic matrix, inorganic fillers, and a silane coupling agent, which connects the two components [7]. The organic matrix consists of several monomers, such as BisGMA (2,2-bis[p-(2’-hidroxy-3’-methacryloxypropoxy)phenyl]- propane), UDMA (urethane dimethacrylate), TEGDMA (trietylenglycol dimethacrylate), DMAEMA (dimethylaminoethyl methacrylate) and various additives (photoinitiators-camphoroquinone, inhibitors, stabilizers). After the polymerization reaction of the monomer mixture of resin composites, an organic matrix as a three-dimensional cross-linked network is formed [8].

Photo-activation is the most used method to initiate the polymerization reaction of resin composites, and several methods are available with their own advantages and disadvantages, regarding the characteristics, final restoration and sustainability of the restored teeth [8]. During the polymerization of light-cured resin composites, monomer conversion is never complete [9] and a percentage of monomer and double bonds remain unreacted as methacrylate pendant groups. This is due to the occurrence of the gel effect, which reduces the diffusion rate of the components of the organic mixture and blocks the polymeric network preventing their total conversion to polymerization [8].

The final properties of photo-cured composites are directly influenced by the degree of conversion of double carbon bonds to single carbon bonds attained during polymerization [10]. The degree of conversion of monomer to polymer in dental composite resins varies between 40% and 75% [9] or between 35% and 77% [7,11,12]. Chung and Greener obtained a degree of conversion of photo-cured composites ranged from 43.5% to 73.8% [13]. The increase of the irradiation time from 30 to 50 s led to a significant decrease of the residual monomer content [14].

Free amounts of monomers have been considered, since 10% of unconverted double bonds can be eluted from composites [9]. The amount of monomers ranges 0.5–2% by weight in water and 70% by weight in ethyl alcohol [11]. More than 30 chemicals were found to be released composites, including residual monomers, as well as initiators, catalysts, metal ions, impurities [15].

Different components can be released from composite restorations in the oral environment and they are associated with hazardous effects such as irritation in the mouth, inflammation, allergic reactions or cytotoxic effects [1,16,17]. The residual unreacted monomer acts as a plasticizer and reduces the mechanical properties of the dental composite [18]. Unreacted monomers trapped in the polymer network may reduce the clinical longevity of the dental composite fillings, as well as their colour stability. However, unreacted monomers are not the only potential toxic compounds. Long-term degradation of the resinous matrix can produce harmful compounds.

Several studies have used scanning electron microscopy (SEM) and Fourier transform infrared (FTIR) spectroscopy to evaluate the surface properties [19,20] and degree of conversion of dental composites [7,11,12,13,18,21]. The determination of the quality and quantity of the residual monomers eluted from polymerized dental composites is usually performed by using high-performance liquid chromatography (HPLC) [22,23,24,25,26] and represents an important step for evaluating material biocompatibility.

Based on these concerns, considerable effort has been made to develop material systems that may have fewer negative biological consequences. Our team designed and developed new resin composite materials with excellent biocompatibility [27,28] and good properties to match the demanding clinical situations represented by cervical dental lesions. The addition in composite of the bioglasses release ions needed for re-mineralisation and hydroxyapatite as bioactive and biocompatible material possessing osteoconductive properties, make the composite susceptible for cervical dental lesions restorations.

The aim of this study was to assess the physico-chemical properties, such as the degree of conversion and residual monomers in cured samples of some dental composites (three commercial and three experimental) using FTIR-ATR (attenuated total reflectance) and HPLC-UV (ultraviolet detection), and to evaluate the mechanical properties of the materials. The null hypothesis was that there would be no differences between experimental and commercial resin composites regarding the evaluated parameters.

## 2. Materials and Methods

### 2.1. Preparation of Composite Specimens

The inorganic filler of experimental resin composites consists of silanizated powders based on mixture of colloidal silica—SiO_2_ (Degussa, Germany), quartz (Uricani, Romania), hydroxyapatite with zirconia (HA-ZrO_2_), BaO and BaF_2_ glasses, obtained by the conventional melting method (UBB-ICCRR laboratory). Surface treatment of all the fillers was made by silane coupling agent, γ-methacriloyloxypropyl-trymethoxysilane (A-174) (Sigma-Aldrich Inc., St. Louis, MO, USA), which gives better interfacial adhesion between inorganic particles and matrix due to the siloxane bonds presented on the surface of inorganic particles.

The organic matrix of experimental resin composites consists of a monomer mixture (presented in Table 1) with camphorquinone (Sigma-Aldrich Inc., St. Louis, MO, USA) (0.5% relative to the liquid mixture)/amine (1%—2-(Dimethylamino)ethyl methacrylate; Sigma-Aldrich Inc., St. Louis, MO, USA) as initiator/activator system. The experimental composites PM, P14M, P2S were prepared as monopastes, by dispersing the silanized bioactive inorganic fillers in the organic matrix. 

For comparison, three commercial composites recommended for the restoration of dental cervical lesions (G-ænial Anterior®, GC EUROPE N.V.; Leuven, Belgium; Enamel plus HRi®, Micerium S.p.A; Avegno, Genoa, Italy; Charisma®, Heraeus Kulzer; Berkshire, UK) (named Ge, En and Ch) were used (presented in Table 1).

### 2.2. FTIR-ATR Spectroscopy

The residual double bonds (RDB) and degree of conversion (DC) of dental composite samples were determined by using Fourier transform infrared spectroscopy in total reduced reflectance (FTIR-ATR) measurements. Specimens with standard dimensions (15 mm in diameter, 1 mm in thickness, n = 10 per composite type) were prepared/obtained from each composite material. The samples were polymerized from both sides in a Teflon mould for 20 s. In order to remove the material excess, a polyester film was placed onto the surface of the mould and pressed with a glass plate The light-curing was carried out with a light-curing unit (Woodpecker LED Curing light, Guilin Woodpecker Medical Instrument Co., Ltd.; Guangxi, China, absorbs light in the 400–500 nm, I ≈ 600 mW·cm^−2^, light guide diameter 8 mm). First, the spectrum of the unpolymerized sample was measured.

The samples were analysed with a FTIR spectrometer (Jasco 610, Jasco International Co., LTD., Tokyo, Japan) in an ATR mode, with a scanning range from 4000 to 550 cm^−1^ at a speed of 4 cm^−1^·s^−1^ and with an average of 128 measurements in the final spectrum. All samples were analysed in a mould. First, the samples were scanned for its FTIR spectrum without being irradiated and then were cured, and, FTIR spectra of the polymerized samples were measured.

As the monomers that make up the organic phase of dental composites are mostly dimethacrylates, an absorption band of 1635–1640 cm^−1^, corresponding to C=C double bonds in the methacrylate groups, was used for the quantitative determination of unreacted methacrylate groups. The peak intensities were compared with an internal standard, the C-C aromatic absorption band at 1608–1610 cm^−1^, which did not participate in the polymerization reaction.

The unreacted methacrylate groups were quantitatively determined by calculating the ratio between absorbance intensities of C=C (aliphatic) and C-C (aromatic) before and after curing using the baseline peak between 1571.7 and 1660.41 cm^−1^ [29]. 

The following equation was used to calculate the ratio of the double-bond content or residual double bonds (RDB) of monomer to polymer in the composite:(1)$$RDB\%=\left(1-\frac{R_{\mathrm{cured}}}{R_{\mathrm{uncured}}}\right)\;\times \;100$$
where *R* is the ratio of aromatic and aliphatic C=C bonds at peak intensities of 1637.27 cm^−1^ and 1608.34 cm^−1^ in cured and uncured composite samples, respectively [29]. Degree of conversion (DC) represents the proportion of polymerized monomers after setting and it is obtained by subtracting the RDB value from 100.

### 2.3. High Performance Liquid Chromatography (HPLC)

The monomer elution was measured using three disc-shaped specimens with standard dimensions (15 mm × 1 mm) from each composite material. The samples were weighed, and the residual monomers were extracted in 25 ml of chloroform for 8 h at the boiling point of chloroform. For the determination of the unreacted monomer remaining in each copolymer, the chloroform extracts (25 ml) were evaporated in vacuum, and the residue was resuspended in 2 ml of acetonitrile, filtered in 0.22 mm PTFE filters and analyzed by HPLC. 

Quantitative analysis was carried out on a HPLC Agilent 1200 series chromatograph (Agilent Technologies, Morge, Switzerland), equipped with G1322A degasser, G1311A quaternary pump, G1329A autosampler, G1315D DAD detector and G1316B TCC SL column thermostat. The chromatographic data were collected and processed using ChemStation software (version B.04.01, Waldbronn, Germany).

Separation was done on a Lichrosorb RP-C18 column (5 μ, 25 × 0.46 cm) (Tracer, Teknokroma, Spain) at room temperature. The mobile phase was composed of a mixture of acetonitrile (A, HPLC grade) and water (B, Milipore ultrapure water) and was applied at a gradient according to the following: 0–15 min, linear gradient 50–80% A; 15–25 min, linear gradient 80–50% A. The mobile phase debit was 0.9 mL·min^−1^ and the sample volume was 20 µL To monitor the elution of the analytes, the DAD detection was accomplished at 195 nm for BisGMA and 203 nm for TEGDMA and UDMA. The compounds were identified by comparison of their retention times with those of the reference compounds under the same HPLC conditions. Every measurement was done in triplicate.

Stock solutions of reference standards (BisGMA, TEGDMA and UDMA from Sigma-Aldrich Chemie GmbH, Steinheim, Germany) 1 mg·mL^−1^) were dissolved in acetonitrile and stored at 4 °C. The ethanol/water solutions with known concentrations of the monomers of interest were used to obtain their calibration curves.

The linearity range of responses of the standards was determined on six concentration levels (0.333–0.0104 mg·ml^−1^) with three injections for each level. The detection limits were 2.9 nmol/ml for BisGMA, 6.2 nmol/ml for TEGDMA and 5.4 nmol/ml for UDMA. Data are presented in μg·mL^−1^.

For the HPLC determination, we used ethanol/water mixture for extraction of monomers. The greater the penetrance of ethanol/water mixture is and the greater their ability to diffuse into the polymer matrix is, the more it allows for leaching compared to the aqueous solution.

### 2.4. Mechanical Properties

#### 2.4.1. Flexural Strength and Flexural Modulus

The flexural strength (FS) and Young’s modulus of bending (YM) were determined in a three-point bending test (*n* = 20) in analogy to ISO 4049:2000 [30]. Twenty samples of each material were prepared (25 × 2 × 2 mm) by using a Teflon mould, and then cured for 20 s according to the manufacturer’s specifications with a halogen lamp (Translux Energy ®/Heraeus-Kulzer) which has a light intensity of 900 mv/cm^2^. The flexural tests were performed using an Universal Testing Machine (LF Plus, LLOYD, Instrument, AmetekInc, West Sussex, England) at a crosshead speed of 0.5 mm/min, with 2 kN loading force. The force measurements, during bending as function of deflection of the beam were done with the universal testing machine. The slope of the linear part of the force-deflection diagram, could be determined the Young’s modulus of bending.

#### 2.4.2. Compressive Strength (CS)

Twenty samples (n = 20) of each investigated material were prepared according to American Dental Association (ADA) Specification no. 27 (cylindrical form samples with 3 mm diameter × 6 mm height). For compressive tests, we used the Universal Testing Machine by applying an axial compressive load at a crosshead speed for 0.05 mm/min.

#### 2.4.3. The Diametral Tensile Strength (DTS)

DTS was determined on specimens with similar cylindrical shape (6 mm in diameter and 3 mm height) in according to American Dental Association (ADA) Specification no. 27. The samples (n = 20) were placed in contact with the supporting plates of the universal testing machine with outside surface of the cylinder, and then a compressive load was applied at a crosshead speed of 0.5 mm/min. The first evaluation of the mechanical measurements was made after the keeping of the specimens for 24 h in the distilled water at 37 °C, and the next evaluation after 3 months of immersion in distilled water (W) or artificial saliva (S).

### 2.5. Data Analysis

Residual double bond (RDB) measures were analysed using a one-way ANOVA independent sample statistical model and Bonferroni post-hoc analysis having as independent variable experimental resin composites (three commercial: Enamel Plus HRI®, Genial anterior®, and Charisma®, and three experimental: PM, P2S, P14M dental composites). Statistical comparisons were considered to be significant at *p* < 0.05. Calibration curves and detection limits, as well as the quantification for studied residual monomers (analyses), were analyzed. Data regarding mechanical properties (FS, CS and DTS) were analysed using a mixed 4 × 2 × 2 ANOVA linked with independent variables. The analysed materials: three commercial (Enamel Plus HRI®, Genial anterior®, Charisma®) and three experimental (PM, P2S, P14M) dental composites were measured independently time variable (initial vs after 3 months) in different storage medium (Saliva vs. Water). Post hoc tests were carried out using Bonferroni correction. Results were considered to be significant at *p* < 0.05. All statistical analyses were performed using IBM SPSS (Windowes, Version 20.0, IBM Corp. Armonk, NY, USA).

## 3. Results

### 3.1. Residual Double Bond Determination

One of the aims of the present study was to characterize the polymerization performance and leaching behavior of the new developed resin composites in comparison with the marketed ones. Thus, residual double bonds (RDB) were evaluated as a prerequisite for polymeric network stability and availability of non-reacted elements. RDB was analyzed by using FTIR-ATR spectroscopy for three commercial (Enamel Plus HRI®, Genial anterior®, and Charisma®) and three experimental (PM, P2S, P14M) dental composites. The monomer conversion to polymer was observed by the decrease, or even the disappearance, of the band corresponding to aliphatic double bond at 1637 cm^−1^ and 1608 cm^−1^ in cured composite sample. Figure 1 presents the comparative FTIR spectra of tested composite samples. 

The RDB values for tested resin composites are presented in Figure 2.

The smallest percentage of RDB was recorded for the experimental composite P2S (13.99%) and the highest one for the commercial resin composite Ch (43.69%) (*p* < 0.05). When comparing commercial materials, the lowest RDB value was recorded for En, which was significantly lower than the RDB values for the other two–Ge and Ch. As for experimental materials, RDB value for P2S was significantly lower than RDB values for the other two experimental resin composites. The results reveal statistically significant lower percentages of RDB in experimental composites, compared with the values obtained for commercial ones [12].

### 3.2. HPLC Analysis of Residual Monomers

The leaching behavior of experimental and commercial resin composites was characterized with HPLC analysis (Figure 3). In Table 2, the HPLC parameters of calibration curves and detection limits as well as the quantification for studied residual monomers (analytes) are presented.

HPLC analyses revealed very close retention times of the components, as well as a very good superposition of chromatogram peaks with the chromatograms peaks of standard BisGMA, TEGDMA and UDMA (Table 2).

The total quantities of the extracted residual monomers by HPLC method are summarized in Table 3. 

BisGMA was released in the highest amount from all composites except En, which released more UDMA and Genial for which no elution was recorded. An increased elution of BisGMA from Ch and PM composites was recorded. From experimental resin composites, PM showed the highest amount of eluted BisGMA, which was three times higher than from P14M resin composite. 

TEGDMA was eluted from all composites, with minimal values recorded for En and Ge. All UDMA based materials eluted UDMA, the highest amounts being recorded for En and Ge.

For both experimental and commercial resin composites, HPLC analysis show that the quantities of residual monomers are less than 1%, except for BisGMA for experimental PM material and Charisma. 

### 3.3. Mechanical Properties

Average values and standard deviations of the compressive strength (CS), diametral tensile strength (DTS), flexural strength (FS) and Young’s modulus (YM), of commercial and experimental dental composites are represented in Figure 4.

Statistical analysis revealed that water and saliva storage induced significant modifications of all mechanical parameters after 3 months for all tested materials, except for a few comparisons for each type of material. Storage medium seemed not to alter the values of mechanical parameters in comparison with the initial ones for: DTS-saliva for Ge and PM, CS-water for Ch, DTS-water and YM-saliva for P14M and YM-water/ saliva for P2S (*p* > 0.05).

Comparing FS initial values, significant differences between all materials were revealed, but after 3 and 15 months, in water, non-significant differences were recorded between Ge-Ch and Ch-PM. Three months of saliva storage was associated with non-significant differences in FS values between Ge-Ch, Ge-PM, Ch-PM and P14M-P2S comparisons.

Enamel has the highest values of FS in all comparisons, except for water storage (when the value for P2S material is slightly higher) and of CS at baseline and after saliva storage. 

Statistical analysis of YM initial values revealed significant differences between tested materials, except for the values for En-PM, Ge-Ch, P14M-PM, as well as for P2S-Ch, PM and En. No statistical differences were obtained when CS initial values of tested materials were compared across material types. Three months water storage resulted in non-significant differences of CS values between En-Ch, En-P2S and Ch-P2S. After 3 months saliva storage, the only non-significant difference in CS values was calculated between Ge-P14M. 

When the initial values of DTS of the tested materials were compared, significant differences were noted for all comparisons excepting En-P14M, En-P2S and P2S-P14M. Three months water and saliva storage induced other more non-significant differences inf DTS values. 

## 4. Discussion

Three experimental resin composites were designed by an autochthonous research institute and were characterized for their leakage behavior, physical-chemical and mechanical properties. Our previous work [27,28,31] focused on physical-chemical (water sorption and solubility) characteristics and on the biological behavior of these materials by observing their cytotoxic effect on stem cells. The ample investigation was carried out in order to improve the composition of the experimental materials and to allow further activities for their clinical implementation. Elaboration of the experimental composites thoroughly focused on their chemical stability and biosafety in order to choose the material providing the best properties.

One of the aims of the present study was to investigate the unreacted double bonds and the quantity of residual monomers for tested resin composites as markers for material stability and clinical behavior. The smallest values of RDB were recorded for experimental composites, followed by commercial resin composites En and Ge. The best behavior was recorded for P2S. Ch showed higher RDB value.

BisGMA was the most eluted compound from composite discs, except for En, which released more UDMA and Ge, which is a BisGMA–free resin composite. All experimental resin composites released less UDMA than commercial materials. En and Ge released more UDMA than the other materials. All types of tested monomers were released in smaller quantities from P2S and P14M materials in comparison with PM material. 

Generally, light-cured methacrylic composites do not have a complete double bond conversion because of the gelation, vitrification, immobilization, and/or steric isolation, thus the identification of BisGMA as an eluted molecule is not surprising. 

BisGMA elution does not seem to follow RDB values, except for Ch, which had the highest values of RDB and BisGMA elution. The release of the monomer can be correlated with the degree of double bonds conversion, but it depends on the measurement technique. By FTIR measurements, RDB does not necessarily need to be correlated with the amount of released residual monomers, because the RDBs detected can remain as pendant groups that could be bound to the polymeric network and are not free to drain [11], which can explain the above mentioned observation. TEGDMA leakage was recorded for all tested materials. As a small low molecular weight monomer, TEGDMA diffuses easily and at a higher rate compared to bulky molecules with a rigid structure such as BisGMA [14,32].

Eluted monomers negatively influence mechanical properties, i.e., decreased wear resistance, hardness, increased tendency of discoloration, and have biological hazardous, biological consequences such as cytotoxic and genotoxic effects on oral cells and tissues [33,34,35,36] and allergic reactions [37]. Residual double bonds suffer oxidation reactions inducing in time chromatic instability [38]. The amount of residual monomers and unreacted double bonds varies according to the methods and conditions of polymerization, as well as the biodegradation of the restorations in the oral cavity, and they are responsible for suboptimal restoration properties [39,40].

Polymerization rate could be partially improved in clinical settings by a strict monitoring of curing protocols. By reducing the polymerization time from 40 s to 20 s it will appear at a higher percentage of unbound molecules susceptible for leaching in oral medium and can affect the pulp tissues. The reduced time could cause differences of approximately 50% [41].

Optimization of the polymerization process could ameliorate these drawbacks, thus improving the physio-chemical and mechanical properties [42] by adding the different type of fillers. Monomer and filler type, filler content, and filler and polymer matrix refractive index have an impact on the ability for light to be transmitted throughout the polymer composite layers [43]. 

The release control of residual monomers and the degree of conversion through different techniques on different materials used in dentistry is in continuous research. New specialized literature presents the techniques for monitoring the release of monomers from the composition of dental materials and their influence on the body [44].

By adding in the inorganic filler, the HA-ZrO_2_ and BaO glasses with high refractive index in the visible region, have an impact on the ability of a complete polymerization of the composite material, resulting in a higher conversion of the monomers.

Our previous study showed that when water and saliva sorption and solubility were investigated, PM material showed the weakest behavior from the three experimental materials [27], which could be associated with the greatest leakage of unbounded monomers and RDB value observed for PM material by the present research. The weakest sorption and solubility behavior of PM material in comparison with the other two experimental composites may be due to the presence of polycaprolactone in PM material [27], which may be degraded by oral microorganisms, inducing instability of the network [45,46]. However, no signs of cytotoxicity of experimental resin composites on stem cells were recorded [27]. 

As for mechanical properties, the highest values for initial FS were obtained for En, followed by P2S, Ge, P14M, PM and Ch and can be explained by the increasing amount of flexible monomers in Enamel, which creates a dense and flexible polymeric structure with an increased elastic deformation of the composite. The high values of flexural strength could also be attributed for the composite with a high filler concentration (P2S material). The most common reason for resin composite restoration failure was restoration fracture [47]. ISO 4049 classified two types of light-curing direct filling resins according to the flexural strength to: type 1- filling for occlusal areas, flexural strength ≥80 MPa and type 2-filling for other indications, flexural strength ≥50 MPa [43]. All tested resin composites have initial FS values above ≥80MPa; water and saliva storage decreased FS under the recommendation limit. 

When appreciating a material for its indication in load-bearing areas, modulus of elasticity has also to be considered. A material with a low modulus of elasticity will result in a higher deformability under high masticatory stress, having catastrophic failures as a final outcome. The stiffness of resin composites was measured by the Young’s modulus, which represents the material’s resistance to elastic deformation. For experimental composites, the highest value of YM was recorded for P14M, which have a filler charge of 79%. Although P2S material has 80% filler content. it has the smallest value of YM from all experimental composites, which, however, was stable in time, in both water and artificial saliva. The smaller the degree of stiffness is, the greater its ability to reduce polymerization shrinkage stress.

Two extreme values of CS were recorded: the highest one for En (344 MPa) and the smallest one for PM (220 MPa). The CS differences between the investigated composites is due to particle size and distribution. The composite Charisma, which has the highest variation in particle size in the composition, shows the slightest variations in compression throughout the range of analysis (3 months) in the immersion of both environments, due to the fact that very small particles inserted take effort in reducing compression fracture incidence.

Water and saliva storage significantly altered CS parameter for almost all composites. At baseline the differences between CS values between tested composites were non-significant, but water and saliva immersion render significant some differences in FS values.

Differences between the values for compression strength between the investigated materials is due to a higher distribution of the particle size of hybrid materials, where very small powder particles inserted between higher particles reduce the interstitial space between them. 

The best results for DTS were recorded by Charisma composite followed by Enamel > P14M > P2S > PM > Genial. As shown, the slightest resistance was recorded for Genial. The composition of the organic phase, the nature and proportion of the monomers, the polymerization system significantly influence the tensile strength of the composites.

## 5. Conclusions

The determination of RDB and the amount of residual monomers in dental composites are of great importance, as these parameters could be regarded as prognostic factors of the behaviour of dental restorations. Fewer residual double bonds were found to be present in experimental resin composites than in commercial ones. Two of the experimental materials showed less than 1% residual monomers, which sustains good polymerization efficiency. Residual double bonds undergo oxidation reactions, which in turn lead to instability color and the release of residual monomers from the mass of composite, causing a decrease in wear resistance of the material and the occurrence of adverse effects in contact with mucous membranes of the mouth.

The in vitro study presented highlights of the release of residual monomers from composite materials and the influence of these monomers on physical and mechanical properties. Experimental resin composites have good mechanical properties, which makes them recommendable for the successful use in load-bearing surfaces of posterior teeth. 

## Figures and Tables

**Figure 1 materials-12-02109-f001:**
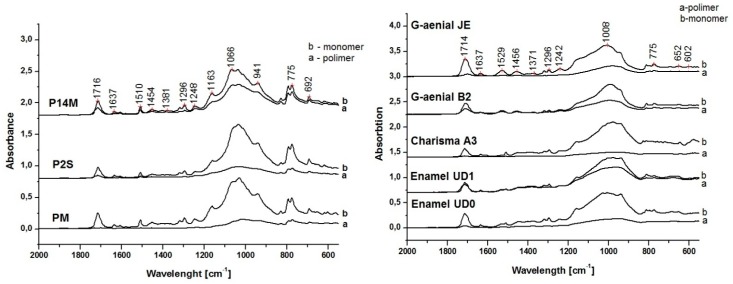
FTIR-ATR spectra of commercial and experimental dental composites for determining the absorbance of *ν(C=C)* and *ν(CH_2_ )* vibrations, before and polymerization.

**Figure 2 materials-12-02109-f002:**
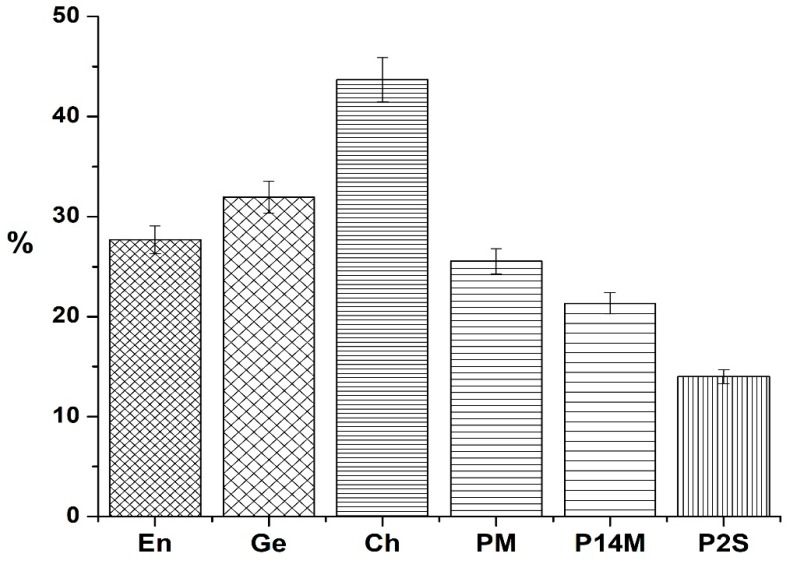
The residual double bonds (RDB) of resin composite discs (En = Enamel Plus HRV; G-ænial Anterior = Ge; Charisma = Ch; PM, P2S, P14M = experimental resin composites).

**Figure 3 materials-12-02109-f003:**
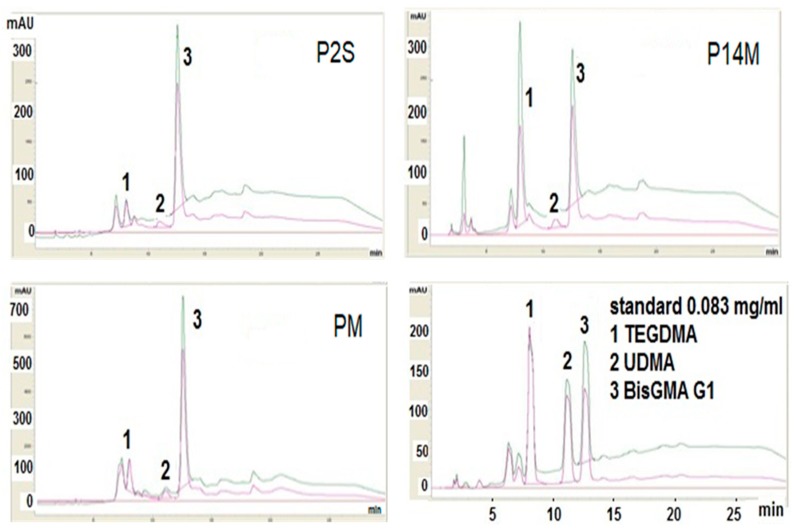
HPLC chromatograms of standards monomers solutions (BisGMA, TEGDMA, UDMA) and representative chromatograms of the extracted monomers in chloroform.

**Figure 4 materials-12-02109-f004:**
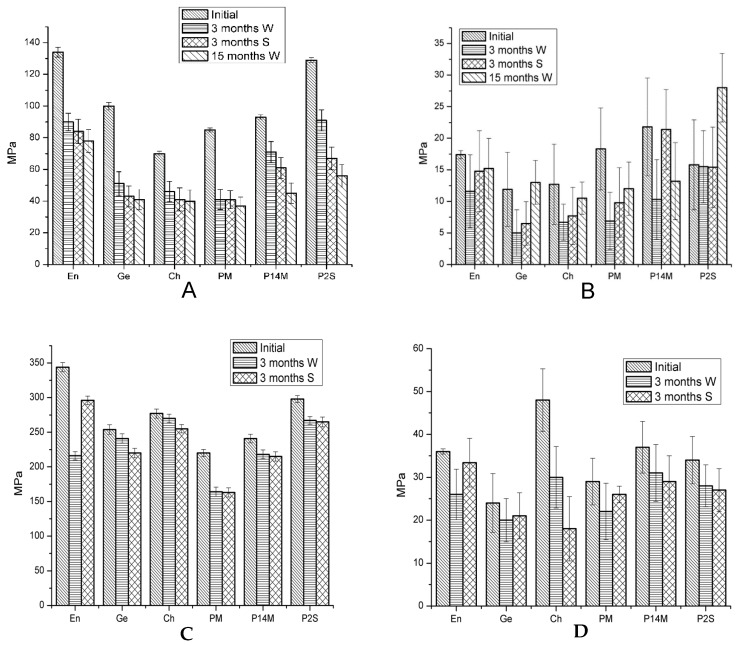
Average values and standard deviations of mechanical properties of flexural strength (FS - graph **A**), Young’s modulus of bending (YM—graph **B**), compressive strength (CS—graph **C** ), diametral tensile strength (DTS—graph **D**) after the polymerization reaction (initial) and after 3, respectively 15 months (only for FS and YM) of immersion in distillated water (W) and artificial saliva (S); En = Enamel plus HRI, Ge = G-ænial anterior, Ch = Charisma.

**Table 1 materials-12-02109-t001:** Composition of experimental and commercial dental restorative composites.

Composite Material	Manufacturer	Composition*
PM	UBB-ICCRR, Cluj-Napoca, Romania	Resins: BisGMA, UDMA, PCL diol and TEGDMA. Fillers: HA-ZrO_2_, silica, BaO glass 78%wt
P14M	UBB-ICCRR, Cluj-Napoca, Romania	Resins: BisGMA, UDMA and TEGDMA. Filler: silica, BaO glass, BaF_2_ glass 79%wt
P2S	UBB-ICCRR, Cluj-Napoca, Romania	Resins: BisGMA, UDMA and TEGDMA. Fillers: HA-ZrO_2_, quartz, silica 80%wt
Enamel plus HRi®	Micerium S.p.A, Avegno GE Italy	Resins: UDMA, BisGMA, 1,4-butandiol-dimethacrylate. Fillers: glass filler, highly dispersed silicone dioxide 53% vol. 75%wt
G-ænial Anterior®	GC EUROPE N.V. Leuven	Resins: UDMA, dimethacrylate co-monomers. Fillers: pre-polimerized fillers containing silica, pre-polimerized particles containing strontium and lanthanoid fluride, silica, fumed silica.
Charisma®	Heraeus Kulzer, NewburyBerkshire, UK	Resins: BisGMA and TEGDMA. Fillers: Ba-Al-B-F-Si Glass, Pyrogenic SiO_2_ 61% vol. 78%wt

Notes: UBB-ICCRR = Babes-Bolyai University, Institute of Chemistry Raluca Ripan, Cluj-Napoca Romania; *BisGMA: 2,2-Bis[p-(2-hydroxy-3-methacryloyloxypropoxy)-phenyl]-propane (UBB-ICCRR); TEGDMA: triethylene glycol dimethacrylate (Sigma-Aldrich); UDMA: urethane dimetacrylate (Sigma-Aldrich); HA-ZrO_2_ hydroxyapatite–zirconia (UBB-ICCRR); PCL diol: polycaprolactone diol (Sigma-Aldrich).

**Table 2 materials-12-02109-t002:** HPLC analysis: regression equations of the calibration curves, retention times and detection limits of the investigated compounds at 205 nm and 275 nm (X = quantity of monomer mg/ml; Y = pick area).

Analyte/Parameters	BisGMA	TEGDMA	UDMA
Regression equation	Y = 62674.3526X − 59.129361	Y = 71611.8675X − 112.45203	Y = 45862.6576X − 101.09324
R^2^	0.99991	0.99993	0.99990
LOD, (μg·mL^−1^)	5.77	5.14	5.84
LOQ, (μg·mL^−1^)	19.24	17.2	19.46
Retention times, (min)	12.581	8.045	11.109

**Table 3 materials-12-02109-t003:** Quantities and rates of the extracted residual monomers in chloroform reported to 1 ml.

Sample	BisGMA G1		TEGDMA		UDMA	
	µg/ml	%	µg/ml	%	µg/ml	%
Enamel Plus	1.13647	0.3	0.046782	0.012	1.66543	0.438
Genial	0	0	0.0114773	0.033	1.72004	0.49
Charisma	3.10570	1.208	0.158634	0.062	0.38482	0.15
PM	3.06525	1.25	0.322001	0.13	0.21718	0.08
P2S	1.30822	0.52	0.150991	0.060	0.115672	0.046
P14 M	1.09551	0.479	0.556638	0.243	0.138829	0.061

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
