# Peer review of "Evaluation of the Degree of Conversion, Residual Monomers and Mechanical Properties of Some Light-Cured Dental Resin Composites"

_materials, 2019, doi:10.3390/ma12132109_

Round 1

Reviewer 1 Report

The article is well written, easy to read and describes the real problem concerning dental resin composites.

Base on the importance of residual monomers and mentioned toxicology I think that it would be useful to specify that the author will not investigate toxicology in this article. The Material and Method section is correctly described. In section 3. Results I would suggest to improve the way of presenting result on Figure2. It would be interesting to see RDB results depending on time, bearing in mind that mechanical results are presented with changes over time. Figure 3 has too low resolution. Finally it will be nice to see any suggested direction or alternative perspectives in Section5. I would suggest to add detail on experimental resin in line 450.

In my opinion the assessment of toxicity should be carried out especially for your composite materials at least pointed out especially in correspondence to RDB. This could consider composition of material, and also the residual monomers. I think that, it would be useful to specify/clarify that you will not investigate toxicology in this article.

The whole manuscript needs a quick edit, there is a few missing or added words, typos or the font was wrong;

line 23 word-WITH is not needed

line 36 recommend-spelling

line 97- missing space between references

line 125- change for-of all the

line 173- change for ml

line 206 -change for kN

line 225- experimental

lines 232-236 This sentence is confusing I would recommend to re-write it: Experimental resin composites (three commercial - Enamel Plus HRI®, Genial anterior®, and Charisma®), three experimental (PM, P2S, P14M) dental composites, storage medium (Saliva vs. Water) were measured independently time  variable (Initial vs After 3 months). Post hoc comparisons were carried out using Bonferroni correction.

Figure 2 - make more clear for the reader,

Figure 3- too low resolution,

line 282- spelling peaks?? 

line 302 Young's

line 345 resulted in...

line 354 physical?

line 412 common/or cause don't need two words

line 421 material's

line 427 recorded

line 435 render

line 446 Fewer

line 450 recommend

lines 551-555- change font

Author Response

Point 1:

Base on the importance of residual monomers and mentioned toxicology I think that it would be useful to specify that the author will not investigate toxicology in this article. 

Response 1:

Te toxicology of this materials was previous investigated on in paper mentioned as reference 27.

Point 2: In section 3. Results I would suggest to improve the way of presenting result on Figure2. It would be interesting to see RDB results depending on time, bearing in mind that mechanical results are presented with changes over time. 

Response 2: We modified the figure 2. When we started the protocol for determining the residual double bond we didn't take into account the behavior in time of this materials and evaluation of RDB values.

Point 3: Figure 3 has too low resolution. 

Response 3: We modified the resolution of figure 3.

Point 4: Finally it will be nice to see any suggested direction or alternative perspectives in Section 5. I would suggest to add detail on experimental resin in line 450.

Response 4: We have mentioned the perspectives of these materials in section 5.

Point 5: In my opinion the assessment of toxicity should be carried out especially for your composite materials at least pointed out especially in correspondence to RDB. This could consider composition of material, and also the residual monomers. I think that, it would be useful to specify/clarify that you will not investigate toxicology in this article.

Response 5: We have mentioned that the toxicity of materials presented in this study was previous investigated and the results published in paper mentioned as reference 27.  

Point 6: The whole manuscript needs a quick edit, there is a few missing or added words, typos or the font was wrong;

line 23 word-WITH is not needed

line 36 recommend-spelling

line 97- missing space between references

line 125- change for-of all the

line 173- change for ml

line 206 -change for kN

line 225- experimental

lines 232-236 This sentence is confusing I would recommend to re-write it: Experimental resin composites (three commercial - Enamel Plus HRI®, Genial anterior®, and Charisma®), three experimental (PM, P2S, P14M) dental composites, storage medium (Saliva vs. Water) were measured independently time  variable (Initial vs After 3 months). Post hoc comparisons were carried out using Bonferroni correction.

Figure 2 - make more clear for the reader,

Figure 3- too low resolution,

line 282- spelling peaks?? 

line 302 Young's

line 345 resulted in...

line 354 physical?

line 412 common/or cause don't need two words

line 421 material's

line 427 recorded

line 435 render

line 446 Fewer

line 450 recommend

lines 551-555- change font

Response 5: We made all the modiffication recommended, point by point. We modified the confused sentence from line 232-236.

Reviewer 2 Report

The authors presented a well elaborated, structured, systematized and written research paper.
The article, in general, is fine. However, it needs minor revisions.
Introduction
The first figure that appears on line 69 has no caption or reference in the text. It is not clear if the figure is illustrative of the introduction or if it is part of a graphic abstract.
line 80 - reference should be made to the authors / studies referring to the above mentioned conversion rate ranges.
Materials and methods
Line 126, 130, 148 - lack town and strips in trademarks.
Line 132 - should refer to table 1
Samples with different sizes were used for each evaluation. Because? Authors should demonstrate the calculation of the sample for each of the evaluation methods.
Results
They are well explained and with explanatory graphics
Discussion
The authors should emphasize the need to develop monomer to polymer conversion tests for the clinically used composite resins since there are in the recent literature systematic reviews on BPA release in composite resins and sealants and their dose / effect relationship in effects systems. (Ref. Doc: 10.3390 / ijerph16091627)

Author Response

Point 1: The first figure that appears on line 69 has no caption or reference in the text. It is not clear if the figure is illustrative of the introduction or if it is part of a graphic abstract.

Response 1: The first figure that appears on line 69 represent graphical abstract of this paper. In the revised paper we deleted them.

Point 2: line 80 - reference should be made to the authors / studies referring to the above mentioned conversion rate ranges.

Response 2: In the line 80 we have mentioned the authors that present in their paper the values of conversion rate ranges.

Point 3: Materials and methods
Line 126, 130, 148 - lack town and strips in trademarks.
Line 132 - should refer to table 1

Response 3: We have mentioned the town and strips in trademarks of the substances from the composition of experimental restorative composites.

Point 4: Samples with different sizes were used for each evaluation. Because? Authors should demonstrate the calculation of the sample for each of the evaluation methods.

Response 4: The dimension of the samples for flexural tests is in according to ISO 4049:1998, and the dimensions aamples for compressive strength and diametral tensile strength according to American Dental Association (ADA) Specification no. 27.

Point 5: Discussion
The authors should emphasize the need to develop monomer to polymer conversion tests for the clinically used composite resins since there are in the recent literature systematic reviews on BPA release in composite resins and sealants and their dose / effect relationship in effects systems. (Ref. Doc: 10.3390 / ijerph16091627)

Response 5: We have mentioned in the discussion the necessity to develop monomer to polymer conversion tests for the clinically used composite resins, regarding the BPA release. We inserted the recommended bibliography.